# Prediction of Successful Pharmacological Cardioversion in Acute Symptomatic Atrial Fibrillation: The Successful Intravenous Cardioversion for Atrial Fibrillation (SIC-AF) Score

**DOI:** 10.3390/jpm12040544

**Published:** 2022-03-30

**Authors:** Jan Daniel Niederdöckl, Alexander Simon, Nina Buchtele, Nikola Schütz, Filippo Cacioppo, Julia Oppenauer, Sophie Gupta, Martin Lutnik, Sebastian Schnaubelt, Alexander Spiel, Dominik Roth, Fritz Wimbauer, Isabel Fegers-Wustrow, Katrin Esefeld, Martin Halle, Jürgen Scharhag, Thomas Laschitz, Harald Herkner, Hans Domanovits, Michael Schwameis

**Affiliations:** 1Department of Emergency Medicine, Medical University of Vienna, 1090 Vienna, Austria; nikola.schuetz@meduniwien.ac.at (N.S.); filippo.cacioppo@meduniwien.ac.at (F.C.); julia.oppenauer@meduniwien.ac.at (J.O.); sophie.gupta@meduniwien.ac.at (S.G.); martin.lutnik@gmail.com (M.L.); sebastian.schnaubelt@meduniwien.ac.at (S.S.); alexander.spiel@gesundheitsverbund.at (A.S.); dominik.roth@meduniwien.ac.at (D.R.); harald.herkner@meduniwien.ac.at (H.H.); hans.domanovits@meduniwien.ac.at (H.D.); michael.schwameis@meduniwien.ac.at (M.S.); 2Zentrale Notaufnahme, Klinik Ottakring, 1160 Vienna, Austria; alexander.simon@gmx.org; 3Department of Clinical Pharmacology, Medical University of Vienna, 1090 Vienna, Austria; nina.buchtele@meduniwien.ac.at; 4Department of Medicine I, Medical University of Vienna, 1090 Vienna, Austria; 5Clinic for Prevention, Rehabilitation, and Sports Medicine, Technical University of Munich-Klinikum Rechts der Isar, 80992 Munich, Germany; fritz.wimbauer@mri.tum.de (F.W.); isabel.fegers-wustrow@mri.tum.de (I.F.-W.); katrin.esefeld@mri.tum.de (K.E.); martin.halle@mri.tum.de (M.H.); 6DZHK (German Centre for Cardiovascular Research), Partner Site Munich Heart Alliance, 80992 Munich, Germany; 7Department of Sports Medicine, Exercise Physiology and Prevention, Institute of Sport Science, Centre for Sport Science and University Sports, University of Vienna, 1150 Vienna, Austria; juergen.scharhag@univie.ac.at; 8Department of Obstetrics and Gynecology, Landesklinikum Korneuburg, 2102 Korneuburg, Austria; thomas.laschitz@korneuburg.lknoe.at

**Keywords:** symptomatic atrial fibrillation, intravenous pharmacological cardioversion, prediction, score, development, validation

## Abstract

Background: Modern personalised medicine requires patient-tailored decisions. This is particularly important when considering pharmacological cardioversion for the acute treatment of haemodynamically stable atrial fibrillation and atrial flutter in a shared decision-making process. We aimed to develop and validate a predictive model to estimate the individual probability of successful pharmacological cardioversion using different intravenous antiarrhythmic agents. Methods: We analysed data from a prospective atrial fibrillation registry comprising 3053 cases of first-detected or recurrent haemodynamically stable, non-permanent, symptomatic atrial fibrillation presenting to an Austrian academic emergency department between January 2012 and December 2017. Using multivariable analysis, a prediction score was developed and externally validated. The clinical utility of the score was assessed using decision curve analysis. Results: A total of 1528 cases were included in the development cohort (median age 69 years, IQR 58–76; 43.9% female), and 1525 cases were included in the validation cohort (median age 68 years, IQR (58–75); 39.5% female). Finally, 421 cases were available for score development and 330 cases for score validation The weighted score included atrial flutter (8 points), duration of symptoms associated with AF (<24 h; 8 points), absence of previous electrical cardioversion (10 points), and the specific intravenous antiarrhythmic drug (amiodarone 10 points, vernakalant 11 points, ibutilide 13 points). The final score, the “Successful Intravenous Cardioversion for Atrial Fibrillation (SIC-AF) score,” showed good calibration (R^2^ = 0.955 and R^2^ = 0.954) and discrimination in both sets (c-indices: 0.68 and 0.66) and net clinical benefit. Conclusions: A predictive model was developed to estimate the success of intravenous pharmacological cardioversion using different antiarrhythmic agents in a cohort of patients with haemodynamically stable, non-permanent, symptomatic atrial fibrillation. External temporal validation confirmed good calibration, discrimination, and clinical usefulness. The SIC-AF score may help patients and physicians jointly decide on the appropriate treatment strategy for acute symptomatic atrial fibrillation. Registration: NCT03272620.

## 1. Introduction

Modern personalised medicine requires decisions tailored to the individual patient [1]. This is particularly important when considering pharmacological cardioversion for the acute treatment of atrial fibrillation and atrial flutter (AF). In emergency and acute care, AF is one of the most common cardiac arrhythmias, accounting for up to 10% of admission diagnoses. This patient volume is associated with increasing economic burdens on the health care system [2,3]. At the same time, emergency department consultations and hospitalisations due to AF are increasing [4,5]. Although management of haemodynamic instability or spontaneous conversion to sinus rhythm is straightforward, the majority of cases provide less clear indications [6,7,8,9]. The primary crucial decision of whether to attempt pharmacological cardioversion, electrical rhythm or rate control already requires an individualised and robust rationale. Further, more holistic benefit–risk considerations should be made by following a decision-making process in addition to balancing the indications and contraindications of each drug [10]. Recently, ESC recommended intravenous use of vernakalant, ibutilide or amiodarone for cardioversion in certain patients [11]. Although the potential hazards of antiarrhythmics are well known, assessing the likelihood of successful pharmacological cardioversion in an individual patient remains a clinical challenge.

Given the increasing prevalence and financial burden of AF, numerous tools have been developed to assess individual patient risk for complications such as stroke, bleeding, death or long-term sequelae of treatment [12,13,14,15]. A tool to assess the individual probability of successful pharmacological cardioversion in an individual patient with symptomatic AF is not currently available. Ideally, however, such information should be incorporated into shared decision making by the patient and the treating physician. Reliable prognostic tools can help patients better assess the advantages, disadvantages, and likelihood of possible outcomes of different treatment options, enabling them to jointly decide with their physician the most appropriate treatment for them. Therefore, the aim of this study was to develop and validate a score for “Successful Intravenous Cardioversion for Atrial Fibrillation (SIC-AF)” to estimate the individual probability of pharmacological cardioversion in patients with haemodynamically stable, non-permanent, acute symptomatic AF.

## 2. Material and Methods

### 2.1. Study Design/Setting

We analysed data from a prospective AF registry that included all adults with AF presenting consecutively to the academic emergency department of the Medical University of Vienna. More than 90,000 patients are treated annually in our outpatient clinic or the associated intensive care unit. Of these, approximately 600 patients with AF are treated each year. Treatment strategies follow current European Society of Cardiology (ESC) guidelines [11] and are based on the patient’s left ventricular function, haemodynamic status, and medical history related to medications, such as anticoagulants, and comorbidities. Drugs used for rate control include beta-blockers, calcium antagonists, and cardiac glycosides. Pharmacological cardioversion is performed using the intravenous antiarrhythmic agents amiodarone, vernakalant or ibutilide. If electrical cardioversion is required, it is performed with synchronised biphasic direct current electrical delivery. The present study was performed in accordance with ICH-GCP recommendations and the Declaration of Helsinki, and was approved by the ethics committee of the Medical University of Vienna (registration number 1568/2014).

### 2.2. AF Registry

Details of the registry have been described previously [16]. Briefly, the registry includes all consecutive cases of AF since January 2011 that were confirmed by 12-lead ECG. Informed consent was obtained prior to enrolment. The study nurses recorded vital signs such as heart rate, blood pressure, and oxygen saturation, symptoms attributable to AF, time of onset of symptoms, type of AF and treatment, including electrolyte replacement, rate-controlling medications, and cardioversion attempts. In addition, demographic data, medical history, concomitant medications, previous attempts at electrical cardioversion, CHA2DS2-VASc score, results of blood gas analysis, blood count, chemistry, standard coagulation tests, thyroid function, troponin and NT-proBNP levels were documented. The registry is registered at clinicaltrials.gov, accessed on 5 September 2017 (NCT03272620).

### 2.3. Prediction Model

Cases of new-onset or recurrent haemodynamically stable, non-permanent symptomatic AF enrolled in the registry between January 2011 and December 2017 were used to develop and validate the predictive model (Figure 1). Patients with AF classified as permanent, patients with spontaneous conversion to sinus rhythm, and those who received only rate control or underwent primary electrical cardioversion were excluded from the analysis. Following the recommendation of Steyerberg and Verguowe [17], we used external temporal validation in a 1:1 sampling ratio after development of the predictive model. Successful intravenous pharmacological cardioversion was defined as restoration of sinus rhythm confirmed by 12-lead ECG during the emergency department visit after intravenous administration of amiodarone, vernakalant or ibutilide without an attempt at electrical cardioversion. Rate control therapy and electrolyte replacement were not considered an attempt at cardioversion.

### 2.4. Statistical Methods

Variables are presented as absolute values (*n*), relative frequencies (*%*) and medians with 25–75% interquartile ranges (IQRs). The Mann–Whitney *U* test (continuous variables) or the chi-square test/Fisher’s exact test (nominal variables) were used for intergroup comparisons. Univariable logistic regression with successful intravenous pharmacological cardioversion (without any attempt at electrical cardioversion) as the dependent variable was performed on available cases. Predictors of successful intravenous pharmacological cardioversion known to be related to both the likelihood of sinus rhythm recurrence and the pathomechanisms of AF or its surrogates were tested [18,19]. Continuous variables in univariable analysis were categorised by selecting clinically relevant cut-off values that were closest to the statistically optimal cut-off values and examined for linear and nonlinear associations. Categorisation of variables yielded parsimony and dichotomy, and cut-off values were optimised for maximum discrimination in the development set. Clinically plausible variables significant in univariable analysis were entered into a multivariable logistic regression model to calculate adjusted odds ratios (ORs) with 95% confidence intervals (95% CIs). A stepwise approach was used, aiming at the most parsimonious model. Interaction was assessed using the likelihood ratio test. For the weighting in the model, these adjusted coefficients of the significant multivariable predictors were inserted as natural integers. The sum of risk scores for each patient was subsequently calculated. Observed versus predicted incidence rates across categories were plotted, and Hosmer–Lemeshow goodness-of-fit tests were used to assess calibration. The discriminative performance of the model was assessed using the c-statistic derived by 1000-fold bootstrapping in both the development and validation sets. To test the robustness of the model, we performed a sensitivity analysis, restricting the observation time from admission to restoration of sinus rhythm to one hour and 30 min. 

Kaplan–Meier estimates for successful intravenous pharmacological cardioversion were calculated across quintiles of the final score (<10, 10–16, 17–20, 21–28 and >28 points).

To evaluate clinical usefulness, decision curve analysis was used. “In this context, the clinical net benefit is the relationship between the benefit of treating those who need treatment and the harm of treating those who do not need treatment. Decision curve analysis allows the evaluation of the clinical net benefit of a predictive tool over a range of threshold probabilities of having a positive outcome. The clinical net benefit is calculated as $\frac{true\;positives}{n}-\frac{false\;positives}{n}\;$ ∗ $\left(\frac{pt}{1-pt}\right)$, where *n* is the total number of patients, and *pt* is the threshold probability of having a positive outcome. True and false-positives are calculated using *pt* as the cut-off point for determining a positive or negative result. This calculation is repeated over a range of clinically meaningful threshold probabilities.” [15]. The development and validation of the prediction model followed the recommendation for such analyses proposed by Steyerberg and Vergouwe [17]. Reporting is based on the TRIPOD statement [19]. Missing data were included as separate categories for each variable as appropriate. For data analysis, we used Stata 17 (StataCorp, College Station, TX, USA). A two-sided *p* value < 0.05 was considered statistically significant.

## 3. Results

### 3.1. Model Development

The development cohort for the predictive model consisted of 1528 cases of first-diagnosed or recurrent symptomatic AF (median age 69 years, IQR 58–76; 43.9% female) enrolled in the registry between January 2012 and June 2014. Of these, 421 cases were finally used for the development of the SIC-AF score (development set). Detailed information on demographic data and patient characteristics can be found in Table 1. An observation time of 8715.9 h was analysed for the development of the prediction model. In the development cohort, the median time from admission to restoration of sinus rhythm was 3.8 h (IQR 2.6–6.0). In total, an intravenous antiarrhythmic drug (amiodarone 208 (49.4%), vernakalant 113 (26.8%), ibutilide 100 (23.8%)) was administered in 421 cases and restored sinus rhythm in 239 cases (56.8%); amiodarone *n* = 89, vernakalant *n* = 74, ibutilide *n* = 76. The corresponding success rates were 42.8%, 65.5% and 76.0%, respectively. The median time to successful intravenous pharmacological cardioversion was 2.5 (IQR 1.6–5.5) hours. Multivariable analysis revealed the following four independent predictors of successful cardioversion (Table 2, Figure 2): the weighted score included atrial flutter (8 points), duration of AF-related symptoms (<24 h; 8 points), absence of previous electrical cardioversion history (10 points), and the specific intravenous antiarrhythmic drug (amiodarone 10 points, vernakalant 11 points, ibutilide 13 points). The cumulative total yielded the “Successful Intravenous Cardioversion for Atrial Fibrillation (SIC-AF) score,” which provided a robust estimate of the success of pharmacological cardioversion. The plotting of the predicted and observed success of intravenous pharmacological cardioversion showed good calibration (*p* = 0.004) (Figure 3). Moreover, the final SIC-AF score showed good discrimination (c-index 0.68; 95% CI 0.65–0.71). The *p* value of the Hosmer–Lemeshow goodness-of-fit test was 0.148. The Kaplan–Meier estimates for successful pharmacological cardioversion in SIC-AF score quintiles (<10, 10–16, 17–20, 21–28 and >28 points) are shown in Figure 4. 

### 3.2. Model Validation

For external validation, we used a cohort of 1525 cases (median age 68 years, IQR 58–75; 39.5% female) included in the registry between June 2014 and December 2017. Of these, 330 cases were used for the validation of the SIC-AF score (validation set). Detailed information on demographics and characteristics is provided in Table 1. The total observation time was 16,866.4 h, and the median duration from admission to recovery of sinus rhythm was 3.4 h (IQR 2.1–5.7).

Intravenous pharmacological cardioversion was attempted in 330 cases (amiodarone 179 (54.2%), vernakalant 80 (24.2%), ibutilide 71 (21.5%). Sinus rhythm was restored in 162 cases (49.1%); amiodarone *n* = 59, vernakalant *n* = 56, ibutilide *n* = 47, corresponding to success rates of 33.0%, 70.0% and 66.2%, respectively. The median time to successful pharmacological cardioversion was 2.8 h (IQR 2.0–7.5). Validation of the SIC-AF score confirmed good calibration (*p* = 0.004, Figure 2) and discrimination (c-index 0.66; 95% CI 0.62–0.71). Sensitivity analyses censoring observation time at one hour (c-index 0.66; 95% CI 0.61–0.71) and at 30 min (c-index 0.66; 95% CI 0.61–0.70) suggested robustness of the model. Decision curve analysis showed a substantial net clinical benefit associated with the use of the SIC-AF score across a wide range of possible thresholds (Figure 5).

## 4. Discussion

In this study, we aimed to develop and validate a prognostic model to estimate the specific probability of success for the use of intravenous antiarrhythmic drugs for pharmacological cardioversion in haemodynamically stable, non-permanent symptomatic AF. Therefore, the SIC-AF score was designed to be as simple as possible for easy application to a real-world cohort of emergency patients with AF. In addition to simplicity, the model was thought to consider that predicting the success of pharmacological cardioversion is most important at baseline. After statistical analysis, which only included immediately available predictors, four independent predictors demonstrated the most parsimonious, robust, and best-performing model: (1) atrial flutter, (2) duration of AF-related symptoms, (3) history of previous electrical cardioversion, and (4) intravenous antiarrhythmic drug.

The SIC-AF score was well calibrated, had good discriminatory power, and decision curve analysis suggested clinically usefulness. However, a final assessment of its clinical usefulness requires further external validation.

### 4.1. SIC-AF Predictors

It is well known that with the duration of AF, the probability of conversion to sinus rhythm decreases [20]. This is thought to be due to a continuous process of electro-anatomical remodelling that promotes heterogeneous conduction, electrical dissociation, and arrhythmic response of the atrial muscle [21,22]. As the disease progresses, structural changes in the atrial myocardium accumulate and promote further arrhythmic episodes and electrophysiological changes, which in turn reduce the likelihood of conversion [23,24,25]. Accordingly, two of the four predictors of the SIC-AF score are surrogates for such remodelling processes. First, the score incorporates information about the history of previous electrical cardioversions, which represents the accumulation of structural atrial changes that may prevent pharmacological cardioversion. Second, fine-grained information on how long the current episode has lasted since onset is a known predictor of the success of pharmacological cardioversion. This can be interpreted as a surrogate for progressive stabilisation of electrical change during an ongoing episode [20]. The third strong predictor that was incorporated into the SIC-AF score was the subclassification of atrial flutter. The observed predictive information inherent in such sub-classification is also supported by previous findings indicating different effects of different agents according to atrial fibrillation or atrial flutter. This may be particularly true for the preferential use of ibutilide in atrial flutter, which may be resistant to other antiarrhythmic drugs. [18] Using this information, the model accounts for the influence of specific indications for different drugs and different response rates. Finally, the fourth predictor included in the SIC-AF score was the particular intravenous antiarrhythmic drug, which allows estimation of the probability of successful intravenous pharmacological cardioversion for the particular agent.

### 4.2. Treatment Efficacy

The model was developed and validated in real-world cohorts of patients with non-permanent symptomatic AF admitted to an emergency department. The importance of targeted, personalised treatment strategies continues to increase in terms of treatment efficacy and management optimisation. Although recent ESC guidelines make clear recommendations for immediate electrical cardioversion in haemodynamically unstable patients, they provide little guidance for intravenous pharmacological cardioversion. However, the group of patients with symptoms that require immediate treatment but do not meet the criteria for haemodynamic instability is large, and given the substantial heterogeneity of this population, the need for directional information is critical. In particular, recent guidelines lack clear recommendations for cardioversion of such patients at several levels. Basically, there is no clear evidence of a long-term benefit of rhythm control over rate control [6,7]. In some patients with AF, early cardioversion or even a primary wait-and-see approach appears to be the correct treatment strategy [9]. Furthermore, there is little robust evidence for the superiority of electrical or pharmacological cardioversion. Here, the required fasting and the risks of the necessary sedo-analgesia must be weighed against the risk of drug-related side effects. Finally, despite the secondary role of amiodarone, there are no clear recommendations for the use of a particular intravenous drug in the same indication/contraindication situation [11]. Therefore, clinical decision making can be very challenging, in part because of the different side effect profiles, but also because of the unclear probability of success of both pharmacological cardioversion, in general, and the use of a specific substance, in particular. An optimal risk–benefit assessment is therefore hardly possible on the basis of previous recommendations alone. A simple tool such as the SIC-AF score can therefore help to support clinical decision making and personalise the treatment of patients with AF who are candidates for pharmacological cardioversion. A simple predictive tool can help personalise treatment for these patients. This would strengthen the shared decision-making process and hasten acute management and potentially reduce the length of stay in crowded emergency departments. Improved patient safety resulting from fewer pharmacologic side effects and improved well-being from earlier treatment success can ultimately reduce the economic burden of AF on emergency departments.

### 4.3. Strengths and Limitations

The greatest strength of this study is the pragmatic study design. As a publicly accessible, multicultural primary care facility of one of the largest tertiary care institutions in Europe, our emergency department has a high caseload and treats a wide range of patients of any socioeconomic status and origin. In patients with haemodynamically stable AF, cardioversion is usually attempted at the earliest opportunity after lab results are available and tentative fluid and/or electrolyte substitution is given to optimise conditions for conversion to sinus rhythm. This likely reflects a common practice in crowded emergency departments with limited space and staff for observation. Given the real-world nature of the study cohort and setting, the consecutive inclusion procedure, and the sample size, the validity of our results can be expected to extend beyond highly selected study populations.

However, the generalisability of our results is limited by the single-centre study design, although we followed the methodological recommendations of Steyerberg and Verguowe and performed robust external temporal validation. Nevertheless, a prospective multicentre validation of the SIC-AF score will be the next step for further evaluation of the promising result of this single centre study. Considering that the primary large cohort size in this study was substantially reduced by excluding patients who received only rate control or primary electrical cardioversion, confirmation of the present results by studies with larger numbers of cases would be essential. Although both the development and validation sets provide statistically robust results, the limitation of the analysis and the potential bias from exclusion should be critically considered. The fundamental risks of bias and confounding in a cohort study also cannot be ruled out and must be kept in mind. The study is also limited by its observational nature, which did not allow standardisation of time to cardioversion attempts and observation periods; thus, potential bias cannot be completely excluded. However, we attempted to compensate for this by sensitivity analyses that restricted the observation time windows from admission to 1 h and to 30 min, and the suggested robustness of the SIC-AF score. Although the results and effect sizes in the main and sensitivity analysis appear consistent, the risk of bias cannot be completely excluded. Above all, the different onset of action of the three investigated substances should be critically reflected. In particular, the mostly delayed effect of amiodarone must be emphasised here. However, further prospective model validation will include predefined monitoring intervals and follow-up periods. Finally, it must be emphasised that not all available intravenous drugs for cardioversion were analysed in the present study. Due to our local standards of care, no suitable data were available for such an analysis. However, considering that some agents, such as class I antiarrhythmics, play an important role in the treatment of acute AF worldwide, they need to be included in future analyses. In addition, potential bias from missing data should be considered. We avoided strict assumptions about missing data and instead included them as a separate category in the models, although this approach cannot exclude bias due to data missing at random.

## 5. Conclusions

With the SIC-AF score, a prognostic model was developed using four clinical parameters to estimate the success of pharmacological cardioversion using three different intravenous antiarrhythmic drugs in a cohort of emergency patients with haemodynamically stable, non-permanent symptomatic AF. External temporal validation confirmed good model calibration, discrimination, and clinical usefulness. Therefore, the SIC-AF score can help patients and physicians in clinical practice jointly decide on the appropriate treatment strategy for acute symptomatic AF.

## Figures and Tables

**Figure 1 jpm-12-00544-f001:**
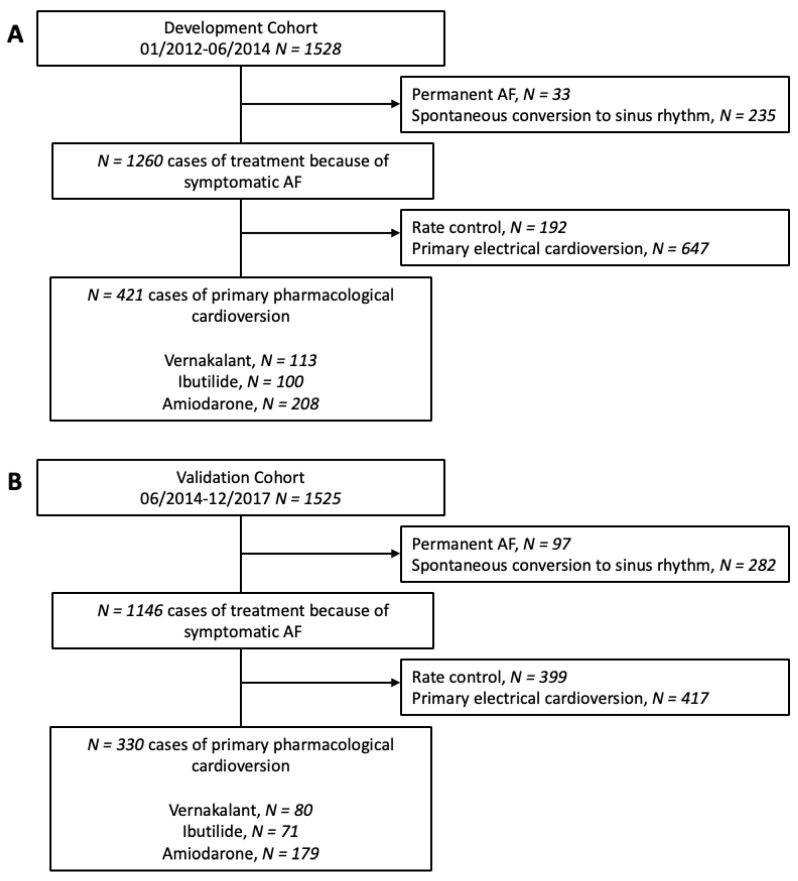
Study flow chart. The development cohort (**A**) consisted of 1528 patients included between January 2012 and June 2014. The 1525 patients in the validation cohort (**B**) were included from June 2014 to December 2017.

**Figure 2 jpm-12-00544-f002:**
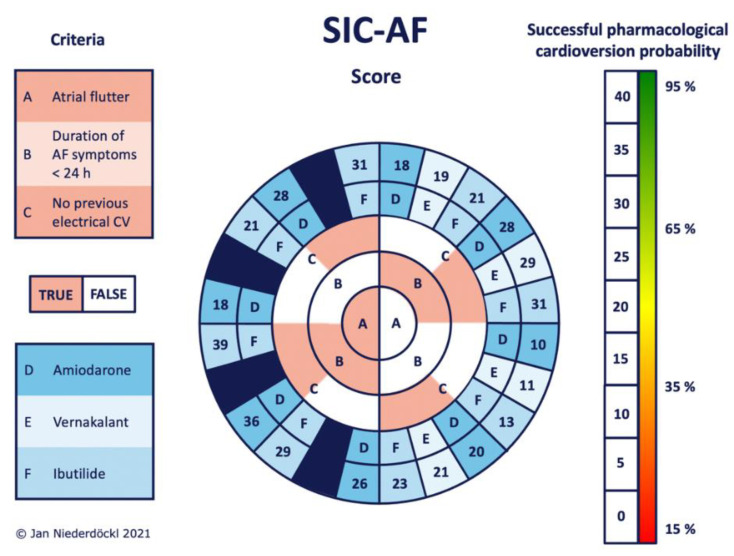
Stratification according to the probability of successful intravenous pharmacological cardioversion using the SIC-AF score. Work along criteria A–C from the middle to the edge. Each corresponding answer leads to the adjacent field of the next circle (reddish = true; white = false). The choice of intravenous antiarrhythmics is included as criterion D, E or F (coded in bluish boxes). The final SIC-AF score can be read directly from the outermost circle. The bar on the right side gives the individual probability of successful intravenous pharmacological cardioversion predicted by the model. Since there is no approval for using vernakalant in atrial flutter, the corresponding fields were excluded (dark blue fields) to avoid misleading information. AF (atrial fibrillation), CV (cardioversion).

**Figure 3 jpm-12-00544-f003:**
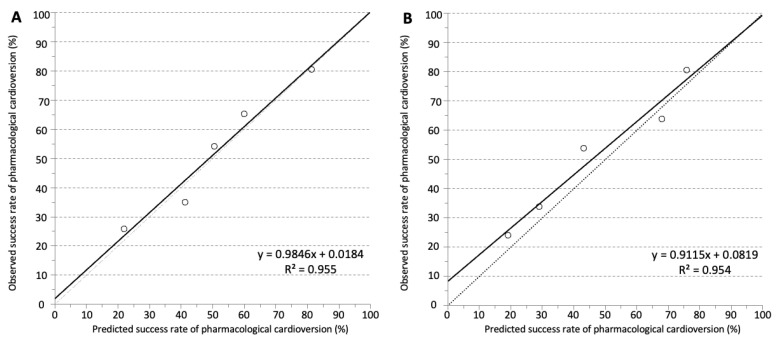
Observed and predicted success rates of intravenous pharmacological cardioversion in (**A**) the development set and (**B**) the validation set. Calibration was visualised by plotting observed vs. predicted incidence rates across quintiles of the SIC-AF score. The dotted line represents perfect calibration. The solid line represents actual calibration.

**Figure 4 jpm-12-00544-f004:**
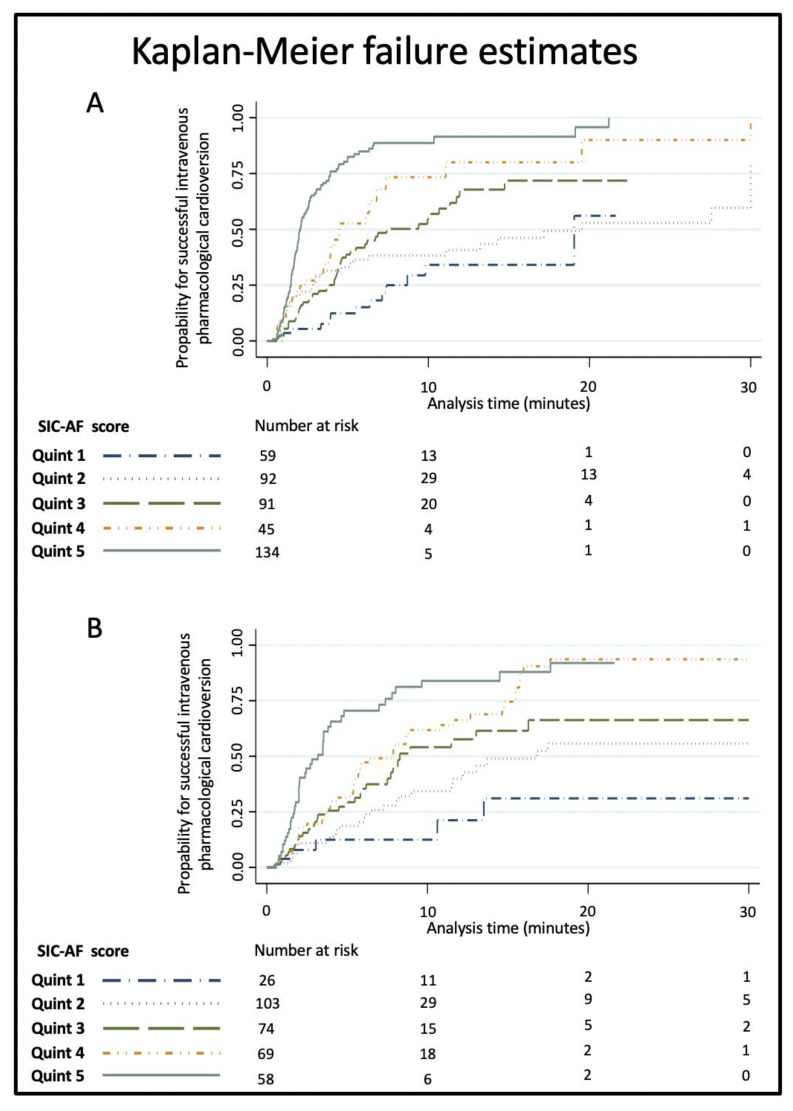
Kaplan–Meier failure estimates for successful intravenous pharmacological cardioversion by SIC-AF score quintiles (<10, 10–16, 17–20, 21–28 and >28 points) in the (**A**) development set and (**B**) the validation set. The probability of cardioversion success increased with increasing quintiles.

**Figure 5 jpm-12-00544-f005:**
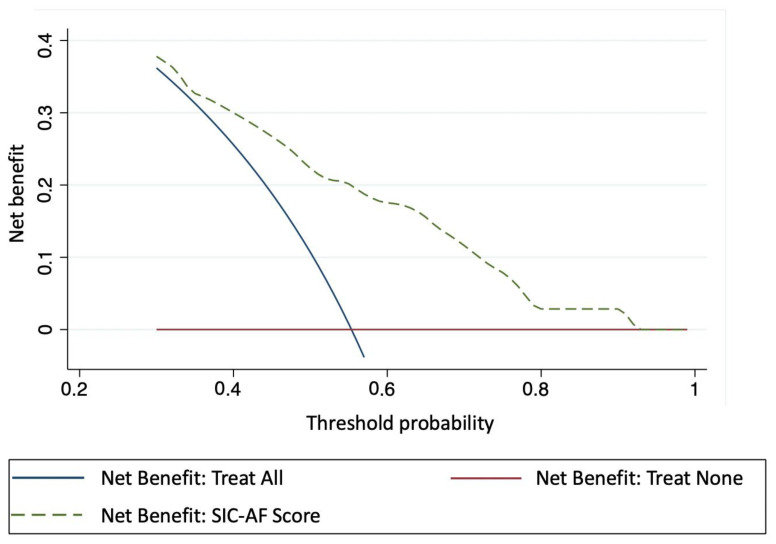
Decision curve analysis showing the clinical usefulness of the SIC-AF score. The X-axis depicts the threshold probability of successful intravenous pharmacological cardioversion. The Y-axis depicts the clinical net benefit of three different strategies: dashed line SIC-AF score; solid blue line: assume all patients would be treated; solid red line: assume no patient would be treated. The SIC-AF score has a positive net benefit across a broad spectrum of threshold probabilities.

**Table 1 jpm-12-00544-t001:** Demographics and baseline characteristics of the development and validation cohorts.

Demographics and Baseline Characteristics		
	Derivation Cohort	Validation Cohort
	*n* = 1260	*n* = 1146
**General characteristics**		
Age, years (IQR)	69 (58–76)	68 (58–75)
Female sex, *n* (%)	553 (43.9)	453 (39.5)
**Comorbidities**		
Heart failure, *n* (%)	131 (10.4)	316(27.6)
Hypertension, *n* (%)	794 (63.0)	663 (57.9)
Diabetes mellitus, *n* (%)	175 (13.9)	176 (15.4)
Prior stroke, *n* (%)	111 (8.8)	77 (6.7)
Coronary artery disease, *n* (%)	223 (17.7)	203 (17.7)
Prior myocardial infarction, *n* (%)	120 (9.5)	93 (8.1)
Peripheral artery disease, *n* (%)	55 (4.4)	49 (4.3)
COPD, *n* (%)	94 (7.5)	108 (9.4)
Valvular disease, *n* (%)	352 (27.9)	267 (23.3)
Current smoker, *n* (%)	101 (8.0)	30 (2.6)
**AF history**		
First AF episode, *n* (%)	182 (14.4)	152 (13.2)
Heart rate, bpm (IQR)	130 (111–146)	127 (102–141)
Atrial flutter, *n* (%)	276 (22)	129 (11)
Duration of AF symptoms, h (IQR)	6 (2–24)	8 (3–24)
Prior electrical cardioversion, *n* (%)	490 (39)	235 (21)
CHA2DS2–VASc (IQR)	3(1–4)	2 (1–4)
**Laboratory**		
Haematocrit, % (IQR)	41(38–45)	42 (38–45)
WBC, G/l (IQR)	8 (7–10)	8 (7–10)
Creatinine, mg/dl (IQR)	1.0 (0.8–1.2)	1.0 (0.9–1.2)
NT–proBNP, pg/mL (IQR)	1160 (409–2883)	1185 (382–2951
hs–Troponin T, ng/l (IQR)	14 (9–26)	15 (8–29)
CRP, mg/dl (IQR)	0.3 (0.1–0.9)	0.4 (0.2–1.3)
INR, (IQR)	1.2 (1.0–2.4)	2.5 (1.7–3.3)
**Treatment**		
Rate control, *n* (%)	192 (15.2)	399 (34.8)
Rhythm control, *n* (%)	1068 (84.8)	747 (65.2)
Electrical cardioversion, *n* (%)	647 (51.4)	417 (36.4)
Vernakalant, *n* (%)	113 (9.0)	80 (7.0)
Ibutilide, *n* (%)	100 (7.9)	71 (6.2)
Amiodarone, *n* (%)	208 (16.)	179 (15.6)

Abbreviations: AF (atrial fibrillation), COPD (chronic obstructive pulmonary disease), CRP (C-reactive protein), hs (high-sensitivity), INR (international normalised ratio), NT-proBNP (N-terminal-pro brain natriuretic peptide), WBC (white blood count).

**Table 2 jpm-12-00544-t002:** Independent predictors of successful intravenous pharmacological cardioversion.

Predictor	Coefficient	95% CI	*p*	Score Points
Atrial flutter	0.82	(0.28–1.35)	0.003	8
Duration of AF symptoms < 24 h	0.83	(0.38–1.38)	<0.001	8
No previous electrical cardioversion	0.98	(0.52–1.45)	<0.001	10
Antiarrhythmic agent				
Amiodarone	Ref			10
Vernakalant	1.13	(0.59–1.67)	<0.001	11
Ibutilide	1.32	(0.74–1.91)	<0.001	13

## Data Availability

The data supporting the results are only accessible to a limited extent and can be requested from corresponing.

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
