# Peer review of "Prediction of Successful Pharmacological Cardioversion in Acute Symptomatic Atrial Fibrillation: The Successful Intravenous Cardioversion for Atrial Fibrillation (SIC-AF) Score"

_jpm, 2022, doi:10.3390/jpm12040544_

Round 1
Reviewer 1 Report
Prediction of successful pharmacological cardioversion in acute symptomatic atrial fibrillation: the Successful Intravenous Cardioversion for Atrial Fibrillation (SIC-AF) score
jpm-1625971
Journal of Personalized Medicine
The authors aimed to develop and validate a predictive model to estimate the probability of successful pharmacological cardioversion using different IV antidysrhythmic agents. They analyzed data from a prospective atrial fibrillation registry of first-detected or recurrent hemodynamically stable, non-permanent, symptomatic atrial fibrillation presenting to an Austrian ED. Using multivariable analysis, they developed and externally validated a prediction score, SIC-AF. In general, the topic is interesting and the study is novel and likely to stimulate further research.
Major issues:
The derivation and validation cohorts are purported to be very large, but the majority of patients were excluded from the analysis. The derivation set actually has n=421 and the validation set has n=330. (Table 1 is misleading). This will substantially limit the results.
Introduction
57- Add additional information and references about how the benefit of cardioversion is unclear. What is the most current consensus? This is the reason why decision-making is integral in these cases.
Material and Methods
133- Who determined “clinical plausibility”? Was a consensus reached?
149- This sensitivity analysis may bias against amiodarone since its effect takes hours. Consider using an observation time that is more typical of an ED visit that involves cardioversion and post-procedural observation (8 hours?)
155- “benefit of treating those who need treatment” will be difficult to define in this population since the benefit of cardioversion has not yet been established. But this should be further clarified, if possible
Results
171- The development set for the predictive model consisted of 421 cases, correct?
183- The predictor “atrial flutter” more specifically should be "absence of atrial flutter"?
216- For external validation, the authors analysed 330 cases, correct?
235- The caption doesn’t match the legend: “solid red line: assume all patients would be treated; solid blue line: assume no patient would be treated.”
Discussion
251- Im not sure that this analysis demonstrates that the score is clinically useful. There needs to be external validation
252- The remainder of the paragraph probably belongs in the introduction
268- May want to comment that atrial flutter is resistant to many drugs, and ibutilide may be preferred in this population (doi:10.1016/j.ijcard.2006.07.017)
Figure 5- Change the label of the bar on the right to “cardioversion probability”. “p” is not defined in the caption
284- There should be greater discussion of How their score recommendations compare to those in the guidelines/current literature?
286 - 291- The majority of this section reads like it should be placed in the introduction.
Limitations
322- The exclusion of patients who received only rate control or underwent primary electrical cardioversion will probably bias the results of the methods. This possibility should be addressed in greater detail
The limitation of small derivation and validation sets should be discussed.
Minor Issues:
53- “atrial” is spelled incorrectly
54- Technically, spontaneous conversion is observed, not treated
60- Consider ordering these with amiodarone last, since ESC describes its “limited and delayed effect” and AHA, Canadian guidelines assign a lower recommendation
99- Correct spelling of “follows”
Figure 3- The Y axes are not labeled
Author Response
Dear Professor Rizzieri and Reviewers,
We would like to thank you for your suggestions and insightful critique, which have enabled us to significantly improve our manuscript "Prediction of successful pharmacological cardioversion in acute symptomatic atrial fibrillation: the Successful Intravenous Cardioversion for Atrial Fibrillation (SIC-AF) score” ( jpm-1625971). In accordance with your suggestions, the manuscript has been revised to include the recommended additions and clarifications. All changes are explained point by point in a response to the reviewers and editors.
We hope this revised manuscript is suitable for publication in the Journal of Personalized Medicine and thank you for your consideration.
In behalf of my coauthors,
Sincerely,
Jan Niederdoeckl, MD, PhD
Reviewer 1
The authors aimed to develop and validate a predictive model to estimate the probability of successful pharmacological cardioversion using different IV antidysrhythmic agents. They analyzed data from a prospective atrial fibrillation registry of first-detected or recurrent hemodynamically stable, non-permanent, symptomatic atrial fibrillation presenting to an Austrian ED. Using multivariable analysis, they developed and externally validated a prediction score, SIC-AF. In general, the topic is interesting and the study is novel and likely to stimulate further research.
Major issues
The derivation and validation cohorts are purported to be very large, but the majority of patients were excluded from the analysis. The derivation set actually has n=421 and the validation set has n=330. (Table 1 is misleading). This will substantially limit the results.
We agree. To avoid misleading readers, the term “large” has been deleted and the number of cases in the sets actually analysed is now stated clearly in the manuscript (including the abstract) (see line 35). We have also addressed this issue in the limitations section (see line 402).
57- Add additional information and references about how the benefit of cardioversion is unclear. What is the most current consensus? This is the reason why decision-making is integral in these cases.
Thank you for this suggestion. We added more references (Pluymaekers et al. 2019; 2019; Roy et al. 2008; Van Gelder et al. 2010) in line 65 and covered this important issue in more detail in the discussion section.
Please see new line 364: In particular, recent guidelines lack clear recommendations for cardioversion of such patients at several levels. Basically, there is no clear evidence of a long-term benefit of rhythm control over rate control (Roy et al. 2008; Van Gelder et al. 2010). In some patients with AF, early cardioversion or even a primary wait-and-see approach appears to be the correct treatment strategy (Kirchhof et al. 2020; Pluymaekers et al. 2019). Furthermore, there is little robust evidence for the superiority of electrical or pharmacological cardioversion. Here, the required fasting and the risks of the necessary sedoanalgesia must be weighed against the risk of drug-related side effects. Finally, despite the secondary role of amiodarone, there are no clear recommendations for the use of a particular intravenous drug in the same indication/contraindication situation. (Hindricks et al. 2020) Therefore, clinical decision-making can be very challenging. On the one hand because of the different side effect profiles, on the other hand because of the unclear probability of success of both pharmacological cardioversion in general and the use of a specific substance in particular. An optimal risk-benefit assessment is therefore hardly possible on the basis of previous recommendations alone. A simple tool such as the SIC-AF score could therefore help to support clinical decision-making and personalise the treatment of patients with AF who are candidates for pharmacological cardioversion.
133- Who determined “clinical plausibility”? Was a consensus reached?
Thank you. The phrase "clinically plausible" is, as you note, inadequate. For the development of the prediction model, variables known to be associated with both the likelihood of sinus rhythm recurrence and the pathomechanisms of AF or its surrogates were tested. (Kafkas et al. 2007; Wijffels et al. 1995). The methods section was adjusted accordingly.
Please see new line 151: Predictors of successful intravenous pharmacological cardioversion known to be related to both the likelihood of sinus rhythm recurrence and the pathomechanisms of AF or its surrogates were tested.
149- This sensitivity analysis may bias against amiodarone since its effect takes hours. Consider using an observation time that is more typical of an ED visit that involves cardioversion and post-procedural observation (8 hours?)
The reviewer is right. Although the results and effect sizes in the main (c-index 0.66; 95% CI 0.62-0.71) and sensitivity analysis censoring observation time at one hour (c-index 0.66; 95% CI 0.61-0.71) and at 30 minutes (c-index 0.66; 95% CI 0.61-0.70) appear consistent, a bias in this respect cannot be completely ruled out.
We have addressed this issue the discussion section:
Please see new line 426: Although the results and effect sizes in the main and sensitivity analysis appear consistent, the risk of bias cannot be completely excluded. Above all, the different onset of action of the three investigated substances should be critically reflected. In particular, the mostly delayed effect of amiodarone must be emphasised here.
155- “benefit of treating those who need treatment” will be difficult to define in this population since the benefit of cardioversion has not yet been established. But this should be further clarified, if possible
Thank you for this suggestion. This section is intended to introduce the reader to the tool of decision curve analysis. We fully agree that without clear evidence of the benefit of cardioversion in general, there will be no benefit for individual methods. However, as long as different methods are used, the benefit is to correctly select the most promising method and thus avoid unnecessary side effects and treatment delays. The decision curve analysis only reflects the potential of the SIC-AF score, which can be seen as a potential advantage over less calculated decisions.
171- The development set for the predictive model consisted of 421 cases, correct?
Yes, that is correct. To avoid confusion among readers from the outset, the number of cases ultimately analysed is now clearly stated in both the abstract and the main text.
We wanted to give the reader a comprehensive, transparent picture of the underlying data. The observation is based on the development cohort (1528 cases) and the validation cohort (1525 cases). 421 cases were finally available for score development and 330 cases for score validation.
Please see new line 196: The development cohort for the predictive model consisted of 1528 cases of first-diagnosed or recurrent symptomatic AF (median age 69 years, IQR 58-76; 43.9% female) enrolled in the registry between January 2012 and June 2014. Of these, 421 cases were finally used for the development of the SIC-AF score (development set).
183- The predictor “atrial flutter” more specifically should be "absence of atrial flutter"?
Thank you for your comment. It could be a misunderstanding. The score assumes that the presence of atrial flutter increases the likelihood of conversion to sinus rhythm, especially if ibutilide is used for cardioversion. The SIC-AF score reflects this fact by adding 8 points.
216- For external validation, the authors analysed 330 cases, correct?
That is correct. The explanation provided above also applies here. To avoid any confusions among readers, the number of cases ultimately analysed is now clearly stated in the abstract and in the main text. We now refer to ”cohorts” (total number of cases available) and “set” (cases finally used for score development and validation). These terms are now used consistently throughout the manuscript.
Please see new line 252: „For external validation, we used a cohort of 1525 cases (median age 68 years, IQR 58-75; 39.5% female) included in the registry between June 2014 and December 2017. Of these, 330 cases were used for the validation of the SIC-AF score (validation set).”
235- The caption doesn’t match the legend: “solid red line: assume all patients would be treated; solid blue line: assume no patient would be treated.”
Thank you for the detailed review. This has been corrected.
251- Im not sure that this analysis demonstrates that the score is clinically useful. There needs to be external validation
The reviewer is right. The wording "The final SIC-AF score was well calibrated, had good discriminatory power, and was clinically useful" is inadequate. We have reworded the sentence.
Please see new line 294: "The SIC-AF score was well calibrated, had good discriminatory power, and decision curve analysis suggested clinically usefulness. However, a final assessment of its clinical usefulness requires further external validation.”
252- The remainder of the paragraph probably belongs in the introduction
Thank you for this suggestion. The paragraph is probably not optimally placed. It was intended as the beginning of the discussion on predictors, to show the relationship between pathomechanisms and predictors. To clarify this, we have moved the text passage in relation to the subheading.
268- May want to comment that atrial flutter is resistant to many drugs, and ibutilide may be preferred in this population (doi:10.1016/j.ijcard.2006.07.017)
Thank you very much for this valuable reference, which we have added to the discussion section.
Please see new line 334: The third strong predictor that was incorporated into the SIC-AF score was the subclassification of atrial flutter. The observed predictive information inherent in such sub-classification is also supported by previous findings indicating different effects of different agents according to atrial fibrillation or atrial flutter. This may be particularly true for the preferential use of ibutilide in atrial flutter, which may be resistant to other antiarrhythmic drugs. (Kafkas et al. 2007).
Figure 5- Change the label of the bar on the right to “cardioversion probability”. “p” is not defined in the caption
Thank you. We have changed the labelling of the bar to “cardioversion probability”.
284- There should be greater discussion of How their score recommendations compare to those in the guidelines/current literature?
This is a valuable suggestion. We have added another paragraph discussing the evidence to date and the specific area of application in which the SIC-AF score or similar predictive models could be optimally used.
Please see new line 364: In particular, recent guidelines lack clear recommendations for cardioversion of such patients at several levels. Basically, there is no clear evidence of a long-term benefit of rhythm control over rate control (Roy et al. 2008; Van Gelder et al. 2010). In some patients with AF, early cardioversion or even a primary wait-and-see approach appears to be the correct treatment strategy (Kirchhof et al. 2020; Pluymaekers et al. 2019). Furthermore, there is little robust evidence for the superiority of electrical or pharmacological cardioversion. Here, the required fasting and the risks of the necessary sedoanalgesia must be weighed against the risk of drug-related side effects. Finally, despite the secondary role of amiodarone, there are no clear recommendations for the use of a particular intravenous drug in the same indication/contraindication situation. (Hindricks et al. 2020) Therefore, clinical decision-making can be very challenging. On the one hand because of the different side effect profiles, on the other hand because of the unclear probability of success of both pharmacological cardioversion in general and the use of a specific substance in particular. An optimal risk-benefit assessment is therefore hardly possible on the basis of previous recommendations alone. A simple tool such as the SIC-AF score could therefore help to support clinical decision-making and personalise the treatment of patients with AF who are candidates for pharmacological cardioversion.
286 - 291- The majority of this section reads like it should be placed in the introduction.
We thank you for this suggestion and have changed the introduction accordingly.
322- The exclusion of patients who received only rate control or underwent primary electrical cardioversion will probably bias the results of the methods. This possibility should be addressed in greater detail
This is an important point. Thank you very much for the discussion on this. We have included this point in a new paragraph in the Limitations section.
Please see new line 403: Considering that the primary large cohort size in this study was substantially reduced by excluding patients who received only rate control or primary electrical cardioversion, confirmation of the present results by studies with larger numbers of cases would be essential. Although both the development and validation sets provide statistically robust results, the limitation of the analysis and the potential bias from exclusion should be critically considered. The fundamental risks of bias and confounding in a cohort study also cannot be ruled out and must be kept in mind.
Minor Issues:
53- “atrial” is spelled incorrectly
Thank you very much. The typo has been corrected.
54- Technically, spontaneous conversion is observed, not treated
Thank you very much. To reflect this, we have changed the wording to "management".
60- Consider ordering these with amiodarone last, since ESC describes its “limited and delayed effect” and AHA, Canadian guidelines assign a lower recommendation
That is a good idea that makes the presentation more consistent. Thank you very much.
99- Correct spelling of “follows”
Thank you for pointing this out. To correct the misunderstanding, we have changed the wording to "study nurses" instead of "study fellows".
Figure 3- The Y axes are not labeled
Thank you very much. We have labelled the Y-axis accordingly.
Reviewer 2 Report
The authors are to be congratulated on performing a pragmatic study on pharmacologic cardioversion (cv) of atrial fibrillation (AF). They were able to validate a prognostic model to assess the probability of success of intravenous pharmacological cv.
Some limitations, such as the single-centre observational character of the study are well adressed by the authors in the discussion section.
Another important point is the selection of drugs in this study and the generalization of results to other institutions and / or other countries. For example, there are several countries where Ibutilide is not available; furthermore; many institutions do not use vernakalant (despite availability, and excellten clinical date) since it is just too expensive. Therefore, it is Flecainide whicht is one of the most often used intravenous drugs for cv of AF in many institutions - at least in patient without structural heart disease, a drug, unfortunately not included in this study. The authory may comment on that. Further prospective model validations should include Flecainide as well.
Nevertheless, the study is importand in demonstrating that development of a cardioversion score may help to "streamline" procedures in a busy emergency department and may add to a pragmatic procedural cv strategy for many patients.
Author Response
Dear Professor Rizzieri and Reviewers,
We would like to thank you for your suggestions and insightful critique, which have enabled us to significantly improve our manuscript "Prediction of successful pharmacological cardioversion in acute symptomatic atrial fibrillation: the Successful Intravenous Cardioversion for Atrial Fibrillation (SIC-AF) score” ( jpm-1625971). In accordance with your suggestions, the manuscript has been revised to include the recommended additions and clarifications. All changes are explained point by point in a response to the reviewers and editors.
We hope this revised manuscript is suitable for publication in the Journal of Personalized Medicine and thank you for your consideration.
In behalf of my coauthors,
Sincerely,
Jan Niederdoeckl, MD, PhD
Reviewer 2
The authors are to be congratulated on performing a pragmatic study on pharmacologic cardioversion (cv) of atrial fibrillation (AF). They were able to validate a prognostic model to assess the probability of success of intravenous pharmacological cv.
Some limitations, such as the single-centre observational character of the study are well adressed by the authors in the discussion section.
Another important point is the selection of drugs in this study and the generalization of results to other institutions and / or other countries. For example, there are several countries where Ibutilide is not available; furthermore; many institutions do not use vernakalant (despite availability, and excellten clinical date) since it is just too expensive. Therefore, it is Flecainide whicht is one of the most often used intravenous drugs for cv of AF in many institutions - at least in patient without structural heart disease, a drug, unfortunately not included in this study. The authory may comment on that. Further prospective model validations should include Flecainide as well.
Thank you very much for the suggestion. We fully agree with the reviewer. Even though we unfortunately cannot make any statements about the antiarrhythmic drugs not routinely used in our centre, we have tried to emphasise this critical aspect when revising the discussion.
Please see new line 431: Finally, it must be emphasised that not all available intravenous drugs for cardioversion were analysed in the present study. Due to our local standards of care, no suitable data were available for such an analysis. However, considering that some agents, such as class I antiarrhythmics, play an important role in the treatment of acute AF worldwide, they need to be included in future analyses.
Nevertheless, the study is importand in demonstrating that development of a cardioversion score may help to "streamline" procedures in a busy emergency department and may add to a pragmatic procedural cv strategy for many patients.